# Neuroprotective and Regenerative Effects of Growth Hormone (GH) in the Embryonic Chicken Cerebral Pallium Exposed to Hypoxic–Ischemic (HI) Injury

**DOI:** 10.3390/ijms23169054

**Published:** 2022-08-13

**Authors:** Juan David Olivares-Hernández, Martha Carranza, Jerusa Elienai Balderas-Márquez, David Epardo, Rosario Baltazar-Lara, José Ávila-Mendoza, Carlos G. Martínez-Moreno, Maricela Luna, Carlos Arámburo

**Affiliations:** Departamento de Neurobiología Celular y Molecular, Instituto de Neurobiología, Campus Juriquilla, Universidad Nacional Autónoma de México, Querétaro 76230, Mexico

**Keywords:** growth hormone, neuroprotection, hypoxia–ischemia, pallium, neurotrophins, synaptogenesis, apoptosis, encephalopathy

## Abstract

Prenatal hypoxic–ischemic (HI) injury inflicts severe damage on the developing brain provoked by a pathophysiological response that leads to neural structural lesions, synaptic loss, and neuronal death, which may result in a high risk of permanent neurological deficits or even newborn decease. It is known that growth hormone (GH) can act as a neurotrophic factor inducing neuroprotection, neurite growth, and synaptogenesis after HI injury. In this study we used the chicken embryo to develop both in vitro and in vivo models of prenatal HI injury in the cerebral pallium, which is the equivalent of brain cortex in mammals, to examine whether GH exerts neuroprotective and regenerative effects in this tissue and the putative mechanisms involved in these actions. For the in vitro experiments, pallial cell cultures obtained from chick embryos were incubated under HI conditions (<5% O_2_, 1 g/L glucose) for 24 h and treated with 10 nM GH, and then collected for analysis. For the in vivo experiments, chicken embryos (ED14) were injected in ovo with GH (2.25 µg), exposed to hypoxia (12% O_2_) for 6 h, and later the pallial tissue was obtained to perform the studies. Results show that GH exerted a clear anti-apoptotic effect and promoted cell survival and proliferation in HI-injured pallial neurons, in both in vitro and in vivo models. Neuroprotective actions of GH were associated with the activation of ERK1/2 and Bcl-2 signaling pathways. Remarkably, GH protected mature neurons that were particularly harmed by HI injury, but was also capable of stimulating neural precursors. In addition, GH stimulated restorative processes such as the number and length of neurite outgrowth and branching in HI-injured pallial neurons, and these effects were blocked by a specific GH antibody, thus indicating a direct action of GH. Furthermore, it was found that the local expression of several synaptogenic markers (NRXN1, NRXN3, GAP-43, and NLG1) and neurotrophic factors (GH, BDNF, NT-3, IGF-1, and BMP4) were increased after GH treatment during HI damage. Together, these results provide novel evidence supporting that GH exerts protective and restorative effects in brain pallium during prenatal HI injury, and these actions could be the result of a joint effect between GH and endogenous neurotrophic factors. Also, they encourage further research on the potential role of GH as a therapeutic complement in HI encephalopathy treatments.

## 1. Introduction

Perinatal hypoxic–ischemic encephalopathy (HIE) is a condition derived from a severe oxygen and energy deprivation, occurring either before or around the time of birth, that activates the appearance of complex molecular, cellular, and physiological alterations that may result in a high risk of permanent neurological deficits or even newborn decease [1,2]. Prenatal hypoxic–ischemic (HI) injury inflicts a devastating damage on the developing brain provoked by a pathophysiological response that leads to neural structural lesions, synaptic loss, and neuronal death [3,4]. Currently, therapeutic hypothermia is the most widely used approach for treating HIE and has been shown to reduce mortality and improve some neurological aspects in injured neonates [5], though a significant proportion of infants so treated still present adverse outcomes including cerebral palsy or cognitive impairments [5,6]. Thus, new treatments, which include the combination of hypothermia with other neuroprotective agents, such as erythropoietin, melatonin, insulin-like growth factor 1, or the use of stem cells, have been explored [5].

Growth hormone (GH) is a protein hormone synthetized and released from the anterior pituitary gland that plays an important role in brain growth, development, and function [7]. Also, GH is expressed locally in the neural tissue and is involved in neuroprotective actions through autocrine/paracrine mechanisms [8]. During embryonic development, GH has important roles in brain maturation, including pro-survival effects [9,10], proliferation and differentiation of neural precursors [11,12,13], neurite outgrowth [14,15], and synaptogenesis [16], among others. Remarkably, recent experimental studies have shown that GH promotes protective and restorative processes in neonatal models of HI injury [17,18]. Also, in vitro observations indicated that GH led to a marked enhancement of endogenous mechanisms of neural repair, reduced apoptosis, and promoted neurite outgrowth and synaptogenesis after HI-injured cerebellar and hippocampal neurons [15,19].

It has been described that HI insult does not result in a uniform global brain injury, but there are brain regions that are more susceptible to oxygen and energy deprivation [20,21]. In preterm infants, HI causes selective damage in diverse brain regions, with the hippocampus and brain cortex (visual and motor) being the most affected. The lesions on hippocampal–cortical circuits have been associated with severe motor and sensory deficits, as well as cognitive impairments that are common in infants that have suffered HIE [22,23,24]. Interestingly, the growth hormone receptor (GHR) expression is upregulated in both the hippocampus and cortical regions after HI insult [25,26], facilitating the neuroprotective action of this hormone [27].

Although the neuroprotective and regenerative effects of GH have been described in the hippocampus in neonatal and adult models [15,17,28,29,30], it is still unclear whether GH exerts similar effects in the cerebral cortex after prenatal HI injury. Since the damage induced during the development of the brain cortex results in severe sensory and motor impairments in infants with HIE, and considering the therapeutic potential of GH on brain repair after neural injury or stroke in both clinical [31,32,33,34,35] and preclinical [36,37,38,39,40,41] studies, it is pertinent to further investigate the possible protective and restorative actions of GH in the brain cortex after prenatal HI injury.

Thus, in the present study, we used the chicken embryo to develop both in vitro and in vivo models of prenatal HI injury in the cerebral pallium, which is the equivalent of brain cortex in mammals [42,43,44], in order to examine whether GH exerts neuroprotective and regenerative effects in this tissue and the putative mechanisms involved in these actions.

## 2. Results

### 2.1. Neuroprotective Effects of GH in Pallial Cell Cultures Exposed to HI Injury

Phase–contrast imaging revealed that, in comparison to normoxia (Figure 1(Aa)), important alterations in the total cell population and general morphology, as well as the number and length of neurites, occurred when the pallial cultures were subjected to the HI insult (Figure 1(Ab)). On the other hand, treatment with 10 nM GH clearly reverted the harmful effects of HI incubation conditions (Figure 1(Ac)). However, the simultaneous addition of a specific antibody against GH seemed to diminish the beneficial effect of GH upon the injured cells (Figure 1(Ad)). Results of the MTT assay (Figure 1B) distinctly show that cell survival decreased significantly under HI conditions (52.63 ± 4.9%, *p* < 0.0001) in comparison to the normoxic controls (102.3 ± 4.7%); whereas the administration of 10 nM GH provoked a significant recovery of cell survival (71.04 ± 4.09%, *p* < 0.05) in the damaged cultures. However, the GH effect on cell survival was neutralized when the cultures were co-incubated with the specific antibody against GH (58.52 ± 3.15%, *p* < 0.05). Furthermore, as shown in Figure 1C, caspase-3 activity increased significantly after HI injury (146.8 ± 7.7%, *p* < 0.001) in comparison with the normoxia group (100 ± 3.4%), whereas the treatment of HI-injured cells with GH decreased it (121.1 ± 9.3%, *p* < 0.05). This effect of GH upon caspase-3 activity was partially abolished by addition of the specific antibody against GH (138.6 ± 2.17%), although this was not statistically different.

### 2.2. Effects of GH upon the Expression of Caspase-8, Bcl-2, ERK1/2 in Pallial Tissue after Exposing Chicken Embryos to HI Injury In Vivo

To verify that the injury conditions to which the chicken embryos were exposed effectively induced hypoxia, we determined the presence of HIF-1α immunoreactivity (IR) by Western blot in the pallial tissue. Appendix A shows that the incubation of chick embryos for 6 h at 12% O_2_ significantly increased HIF-1α IR (130.4 ± 15.2%, *p* < 0.05), in comparison with controls maintained under normoxic conditions (100 ± 3.88%).

Figure 2 shows the results obtained when chicken embryos were subjected to either normoxic, HI, or HI + GH treatments, and their effects upon the expression of several markers involved in the regulation of apoptosis and cell survival (caspase-8, Bcl-2 and the phosphorylation ratio of ERK1/2) were analyzed by Western blot (WB) in the pallial tissue. As described in Figure 2A, caspase-8 IR significantly increased after HI injury (129.2 ± 6.6%, *p* < 0.01) in comparison with the normoxic control (100 ± 5.6%), whereas the administration of GH during HI reverted that effect and returned caspase-8 IR levels to those observed under normoxia (104 ± 5%, *p* < 0.05). Figure 2B shows that, in relation to the normoxia control (100 ± 7.4%), the antiapoptotic Bcl-2 IR band was significantly decreased in the HI group (64.3 ± 4.5%, *p* < 0.01), but it was then importantly increased with GH treatment (143.3 ± 16.2%) as compared with the injured group that received only the vehicle (*p* < 0.001). On the other hand, the exposure to HI insult provoked a significant decrease in the ratio of phosphorylated ERK1/2/ERK (46.7 ± 4.7%, *p* < 0.001; Figure 2C) in comparison with the normoxic control (100 ± 6.5%), whereas the addition of GH treatment in the injured group significantly increased the ratio (68.4 ± 8.4%, *p* < 0.05).

### 2.3. GH Increases Doublecortin IR and NeuN IR in Pallial Cultures Exposed to HI Injury

The effects of GH treatment upon neuronal subpopulations in primary pallial cell cultures exposed to HI conditions were analyzed by determining the presence of doublecortin (DCX, biomarker for neuronal precursors) and neuronal nuclear antigen (NeuN, biomarker for mature neurons) immunoreactivities by immunofluorescence (Figure 3A) and WB analyses (Figure 3C,D).

First, immunofluorescence image analysis shows that HI injury provoked a drastic reduction in the number of DAPI+ cells (Figure 3(Ab)) in comparison with control cultures incubated under normoxic conditions (Figure 3(Aa)), whereas GH treatment significantly reversed this effect (Figure 3(Ac)). This was quantified in Figure 3B, which shows that, in comparison with normoxia (200 ± 11.61 cells/field) the cultures exposed to HI injury suffered a strong decrease in number (154.9 ± 10.09 cells/field, *p* < 0.05); however, a significant increase in the DAPI+ cell number was observed for the treatment with GH (204.3 ± 9.89 cells/field, *p* < 0.05), which recovered levels similar to those in the normoxia condition.

Likewise, the morphology of DCX IR cells and the number of NeuN IR cells also showed important changes between treatments. As is apparent in Figure 3(Ab), the deleterious effect of the HI insult was more evident upon mature neurons, whereas treating the injured cells with GH prevented the damage provoked by HI and stimulated the survival of both DCX IR and NeuN IR neurons (Figure 3(Ac)). These effects of GH during HI conditions upon DCX IR and NeuN IR were confirmed by WB analysis. As shown in Figure 3D, exposure of cultures to HI injury resulted in an important decline in the NeuN IR band (59.30 ± 15.28%, *p* < 0.05) in relation to the normoxic control (100.7 ± 1.45%), but again, the addition of GH during HI conditions significantly constrained such effects and restored the expression of NeuN IR levels (138.5 ± 7.99%, *p* < 0.001), even above those observed in normoxia (*p* < 0.001). On the other hand, Figure 3C shows that the incubation under HI conditions did not modify the presence of DCX IR (116.5 ± 7.64%) when compared with normoxia (100 ± 7%). However, the administration of GH resulted in a significant increase for DCX IR (141.2 ± 17.01, *p* < 0.01) in comparison with both the normoxia and HI-injured controls, indicating a stimulation of this neuronal precursor marker in these conditions.

### 2.4. GH Promotes Cell Proliferation Both In Vitro and In Vivo after HI Injury

The effects of HI injury and GH treatment upon cell proliferation were evaluated, both in vitro and in vivo, using a BrdU labeling assay. The protocol shown in Figure 4A was followed for the in vitro experiment: pallial cultures were incubated under HI conditions, either alone or with the addition of 10 nM GH, in the presence of BrdU for 24 h. At the end of this period proliferating cells that incorporated BrdU (BrdU+ cells) were immunolabeled and quantified under the microscope. Immunofluorescence analysis (Figure 4B) shows the effects of the different experimental conditions upon the proportion of proliferating BrdU+ cells. Interestingly, although the number of total cells (DAPI+) decreased in the cultures exposed to HI harm in comparison with the control, there was no change in the proportion of BrdU+ cells (Figure 4(Bb)), compared with normoxia (Figure 4B); in contrast, GH treatment to the HI cultures increased the number of DAPI+ cells and the proportion of BrdU+ cells (Figure 4(Bc)). Cell quantification (Figure 4C) confirmed these observations, showing that there was no difference in the proportion of proliferating cells between normoxia (39.7 ± 2%) and HI injury (41.6 ± 4.6%), whereas GH treatment provoked a significant increase in BrdU+ cells (51.4 ± 2.4%, *p* < 0.05).

For the in vivo experiments, the protocol depicted in Figure 4D was followed: chick embryos (ED14) were microinjected in ovo with GH and BrdU and incubated under HI conditions for 6 h. After this period, embryos were injected again with GH, incubated under normoxic conditions during an additional 24 h, and then sacrificed for analysis. BrdU+ cells were immunodetected in the hyperpallium region of the chick brain (Figure 4E). As shown in Figure 4F, immunofluorescence analysis reveals that a relevant change occurred in the number of BrdU+ cells when the embryos were exposed to HI (Figure 4(Fb)), compared with the normoxic group (Figure 4(Fa)). Consistent with the in vitro results, a significant increase in the number of proliferating cells was clearly observed in the HI + GH group (Figure 4(Fc)). Figure 4(Fd) is a negative control showing no signal when the BrdU primary antibody was not added during the immunohistochemistry. Results of the quantification of BrdU+ cells in the three experimental groups are shown in Figure 4G. The exposure of embryos to HI insult caused an increase in the percentage of BrdU+ cells (6.27 ± 0.2%, *p* < 0.05), compared with the normoxia group (5.56 ± 0.2%). Interestingly, GH treatment resulted in a significant increase in the proportion of BrdU+ cells (7.18 ± 0.28%) compared with both the normoxia group (*p* < 0.0001) and the HI group (*p* < 0.05), indicating that the hormone stimulated cell proliferation when administered during and after the hypoxia injury.

### 2.5. GH Promotes Neurite Outgrowths in Pallial Cell Cultures after HI Injury

To analyze whether GH was capable of promoting structural plasticity in the HI-injured neurons, cultured neurons from the chicken pallium were immunostained with beta-III-tubulin, and several morphometric parameters such as the number of neurites (Figure 5B), dendritic length (Figure 5C), and dendritic branching (Figure 5D) were evaluated. The image analysis (Figure 5A) shows that HI injury provoked a drastic reduction in the length of neuronal projections (Figure 5(Ab,Af)), compared with that observed in the normoxia control (Figure 5(Aa,Ae). However, the GH treatment clearly reversed this harmful effect (Figure 5(Ac,Ag)). Interestingly, co-incubation with the specific anti-GH antibody neutralized the protective effect induced by GH (Figure 5(Ad,Ah)). Figure 5B shows that in comparison with normoxia (10.31 ± 0.7), the number of neurites was significantly reduced under HI conditions (5.51 ± 0.4, *p* < 0.0001); however, the administration of GH during HI stimulated an important increase in the number of neurites (12.45 ± 0.9, *p* < 0.001), even above normoxic levels. The protective effect of GH practically disappeared when the neutralizing anti-GH antibody was added (7.35 ± 0.70, *p* < 0.001). Consistent with these results, Figure 5C shows that the dendritic length was also drastically reduced in the HI group (98.54 ± 6.88 µm, *p* < 0.0001) in comparison with the normoxia control (238.8 ± 13.19 µm). Remarkably, GH treatment abolished the effect of HI injury by significantly increasing the dendritic length (273.6 ± 22.81 µm, *p* < 0.0001). Then again, the stimulating effect observed with GH during HI insult vanished when the anti-GH antibody was present in the culture (121.5 ± 12.51 µm, *p* < 0.0001). The branch order analysis revealed that, although HI damage significantly decreased the neuronal branching, GH specifically increased dendrites of the first and second order (*p* < 0.0001 and *p* < 0.0001, respectively; Figure 5D). The plasticity effect of GH was also abolished by co-incubating with the antibody against GH (*p* < 0.001 and *p* < 0.001, respectively).

### 2.6. GH Promotes the Expression of Synaptogenic Markers in Pallial Cell Cultures after HI Injury

The effects of HI injury and GH treatment upon the expression of several specific pre- and postsynaptic markers in cell cultures from the embryonic chicken pallium were analyzed by qPCR (Figure 6). It was found that when the cultures were exposed to HI insult the expression of NRXN1, NRXN3, and GAP-43 mRNAs was not different from the levels observed under normoxic conditions; however, they were significantly increased by the addition of GH (3-fold, *p* < 0.05, Figure 6A; 5.2-fold, *p* < 0.05, Figure 6B; and 4.1-fold, *p* < 0.001, Figure 6C). On the other hand, NLG1 mRNA expression (Figure 6D) was reduced after HI (0.4-fold, *p* < 0.05), whereas GH treatment significantly upregulated its expression (2.6-fold, *p* < 0.0001).

### 2.7. GH Regulates the Expression of Local GH, BDNF, IGF-1, NT-3, BMP4, and GHR mRNAs in Pallial Cultures after HI Injury

The local expression of several endogenous neurotrophins and growth factors in response to the different treatments was analyzed by qPCR. Results show that GH, BDNF, IGF-1, NT-3, and BMP4 mRNA expression did not appreciably differ between HI injury and normoxia conditions. However, the treatment with GH during HI conditions significantly increased GH (4.1-fold, *p* < 0.05; Figure 7A), BDNF (9.3-fold, *p* < 0.001; Figure 7B), IGF-1 (6.2-fold, *p* < 0.001; Figure 7C), NT-3 (8-fold, *p* < 0.001; Figure 7D), and BMP4 (3.4-fold, *p* < 0.0001; Figure 7E) mRNA expression in relation to both normoxic and HI-injured cultures. Lastly, GHR mRNA expression (Figure 7F) diminished in response to HI in comparison with the normal control, and was significantly upregulated under HI + GH conditions (4.3-fold, *p* < 0.001) in relation to the other groups.

## 3. Discussion

The chick embryo is a good animal model to analyze the effects of prenatal hypoxia on brain development, as well as the endogenous response to HI damage, without the intervention of maternal and/or placental influences. In addition, the facility to manipulate the eggs and to administer diverse compounds or drugs through the allantoid sac and investigate their therapeutic potential to revert harm, make chick embryos a suitable model for this kind of study.

Here, we aimed to examine the neuroprotective and restorative effects of GH after damage of the avian cerebral pallium—which is analogous to the mammalian brain cortex [42,43,44]—by employing both in vitro and in vivo approaches to induce experimental prenatal HI injury in this model. We found that GH treatment had a clear anti-apoptotic effect that promoted survival of HI-injured pallial neurons, and these actions were associated with the activation of ERK1/2 and Bcl-2 signaling pathways. The results also show that, in cultures, GH favorably protected mature neurons, which were the most affected by HI injury, although it also increased neural precursors. Furthermore, GH promoted cell proliferation in both in vitro and in vivo HI injury models. In addition, GH stimulated regenerative processes such as neurite outgrowth and synaptogenesis in HI-injured pallial neurons. We also demonstrated that the expression of endogenous neurotrophic factors, e.g., GH, BDNF, NT-3, IGF-1, and BMP4, were increased after GH treatment during HI harm, suggesting that the effects observed could be the result of a joint effect between exogenous GH and endogenous neurotrophic factors. Together, these results provide novel evidence supporting the notion that GH promotes neuroprotective and neurorestorative effects in the developing brain cortex after HI injury.

Hypoxic–ischemic encephalopathy does not result in a uniform or global brain injury but causes selective damage to different brain structures that depends on the severity and duration of the insult as well as on the developmental stage of the brain when it occurs. Evidence from clinical practice supported by MRI suggests that in preterm infants (<32 weeks of gestation), HI insult affects motor, sensory, and cognitive functions [20,21,22]. At this stage, the spinal cord, basal ganglia, thalamus, cerebral cortex, and hippocampus are the most frequently damaged brain regions after HI injury [45,46,47,48]. The harm inflicted on these regions may explain the clinical manifestations observed in HI-lesioned infants. Particularly, hippocampal and neocortical circuits are among the most vulnerable regions to HI encephalopathy [49,50,51]. This vulnerability of hippocampal–cortical circuit is likely to be a consequence of the disproportionate activity of excitatory synapses [20], making these cells more susceptible to excitotoxic damage. Although the anti-apoptotic and restorative effects of GH have been amply characterized in the hippocampus after HI injury, both in vitro and in vivo [15,17,18,26,29,30,31], the number of studies regarding the protective and regenerative effects of GH in the brain cortex after prenatal HI injury are limited. Here, we demonstrate that GH is capable of significantly increasing cell survival in HI-injured pallial neurons. This result is in line with previous studies in brains of neonatal rats [52], in cerebellar [19,53] and hippocampal cell cultures [15] exposed to HI injury, as well as in avian and reptilian neuroretinal cell cultures exposed to excitotoxic damage [16,54,55,56]. The mechanisms by which GH promotes cell survival in response to an HI insult have been associated with anti-apoptotic actions. Accordingly, we observed that caspase-3 activity was clearly increased during HI injury, and then significantly reduced by the addition of GH during the insult. These effects on the pallial cultures are similar to those reported previously for this hormone in the cerebellum [19,53], neural retina [54,55,56,57], and hippocampus [15,58].

HIF-1α is a transcription factor that can be expressed rapidly in response to hypoxia during brain ischemia [59]. It was shown here that HIF-1α immunoreactivity was significantly increased in the pallial tissue after the chick embryos were exposed to the HI insult. This finding is consistent with previous reports where HIF-1α was augmented in embryonic chicken cerebellar cultures [19] and the brain cortex [60] after hypoxic–ischemic injury. During hypoxia, HIF-1α is stabilized and translocated into the nucleus where it forms a functional HIF transcription factor that binds to the hypoxia-response elements (HRE) and induces the expression of several genes (v.gr. *Vegf, Epo, iNos, SLC2A 1*, among others) involved in an adaptive response that regulate cell survival, neuroprotection, and regulation of oxygen homeostasis and angiogenesis during ischemic injury [61,62,63,64,65]. Also, it has been reported that HIF-1α attenuates brain damage, modulating the inflammatory response and glial activity [66]. HIF-1α transcription, stabilization, and half-life is also regulated by growth factors, cytokines, and hormones through several pro-survival pathways, such as ERK/MAPK, JAK/STAT, and PI3K/Akt/mTOR [67]. On the other hand, it is also known that GH activates these signaling pathways, either directly or indirectly [17,18,19,53,68]. Therefore, it is possible that during a hypoxic insult, GH (which has been shown to exert antiapoptotic, proliferative, neuroprotective, and anti-inflammatory actions) can promote and potentiate the expression and stabilization of HIF-1α, as well as the effects of this transcription factor on triggering endogenous responses that result in cell survival and the regulation of oxygen homeostasis and energetic metabolism. However, the potential confluent interactions of GH signaling and HIF-1α signaling during HI injury deserve further investigation.

The mechanisms by which GH could exert its neuroprotective effects have been associated with the triggering of the PI3K/Akt, Bcl-2, and ERK1/2 pathways [18,19,53]. Only a few studies have explored the signaling pathways involved in the pro-survival and protective effects of GH in the brain cortex after prenatal HI injury. Here, we observed that the exposure of ED14 chick embryos to hypoxia conditions significantly increased caspase-8 IR in the pallium, but GH treatment reduced the expression of this enzyme to levels similar to those observed in the normoxic controls. Since caspase-8 is the most upstream protease that participates in the activation cascade responsible for death-receptor-induced cell death [69], this result indicates that GH treatment exerts an antiapoptotic effect during prenatal HI injury. On the other hand, it has been described that Bcl-2, an anti-apoptotic protein [70], was upregulated in neurons, glia, and endothelial cells surrounding the infarcted cortex region following focal ischemia [71,72]. We found that Bcl-2 IR was significantly decreased after HI injury; however, GH treatment increased it importantly when administered to the lesioned animals. This result agrees with those reported previously in the hypoxic cerebellum [19,53] or in the retina exposed to excitotoxic damage [73], suggesting that GH induces Bcl-2 as a mediator of its anti-apoptotic effects. The activation of the ERK1/2 signaling pathway is recognized as an adaptive response in neurons displaying signs of damage after HI insult in diverse models [74]. In this study, we show that ERK1/2 phosphorylation is significantly reduced during the HI insult, but then is increased by GH treatment in the damaged pallium. It is therefore possible that the GH-induced neuroprotection observed in this model was mediated, in part, through ERK signaling pathway activation.

As mentioned above, HI affects selected brain regions during prenatal/postnatal stages, depending on the severity and duration of the insult as well as on the developmental stage and state of neuronal differentiation [20,75]. Thus, we examined the effect of GH upon the immature and mature neuronal subpopulations in primary pallial cultures exposed to HI injury. For this we used two neural biomarkers: DCX, to identify neural progenitor cells [76,77], and NeuN, to characterize mature neurons [78]. Our results, using immunofluorescence and WB, show that NeuN IR was importantly reduced after HI insult, whereas DCX-IR did not change. The high susceptibility of the pallial mature neurons to HI injury could be related to the excitotoxicity derived from the activity of glutamate receptors, including N-methyl-D-aspartate (NMDA), α-amino-3-hydroxy-5-methyl-4-isoxazole (AMPA), and kainate (KA), which are highly expressed in mature cortical neurons [79]. It has been described that during HI insult extracellular glutamate is elevated, provoking an exacerbated activation of NMDA, AMPA, and KA receptors that leads to a massive calcium influx into neurons. An increase in calcium concentration results in the activation of protein kinases and other downstream calcium-dependent enzymes that destroy important molecules, disintegrate the cell membrane, induce the release of free radicals from damaged mitochondria, and provoke subsequent cell death [80,81,82]. Remarkably, we observed a significant increment in both NeuN IR and DCX IR biomarkers when the injured primary pallial cultures were treated with GH. These findings are consistent with previous reports describing that GH has protective effects on mature neurons [19,83,84]. It is possible that the mechanism through which GH protected mature neurons could be similar to that described in the neural retina exposed to kainate-induced excitotoxic damage [56,73]. However, the association between GH neuroprotection during excitotoxic damage and that observed in response to HI insult deserves further investigation.

Another possible alternative to explain the increase in DCX IR and NeuN IR after GH treatment during HI injury, could be linked to the role of this hormone upon proliferation and generation of neural cells in a stroke model [30]. To analyze this hypothesis, we used the BrdU paradigm to assess proliferating cells in both in vitro and in vivo models. An increase in the number of BrdU+ cells was observed in the pallial cell cultures after HI injury. Likewise, the in vivo analysis confirmed this finding, and showed two important results: first, that hypoxia injury alone increased the number of BrdU+ cells in the chicken hyperpallium, which agrees with previous reports demonstrating that hypoxia stimulated neural stem cell (NSC) proliferation in regions such as the striatum, hippocampus, and cortex [85,86,87,88,89], as a mechanism of endogenous brain plasticity, and second, that GH was able to significantly increase the number of BrdU+ cells in the chick embryo hyperpallium after HI damage. An interpretation could be that the new proliferating cells have differentiated into mature neurons, as suggested by Sanchez-Bezanilla et al. [30] who reported that GH treatment promoted cell proliferation within the peri-infarcted regions in experimentally stroked mice. Moreover, they tracked the fate of these proliferating cells and found an increase in the number of NeuN- and DCX-positive cells [30]. The ability of GH to promote neurogenesis within the harmed region after stroke is a critical finding, as indicated by previous studies that reported an association between functional motor recovery and the number of newly born neurons in the motor and somatosensory cortex after ischemic injury [90].

Brain structural and synaptic plasticity are two processes that play a relevant role in brain repair after ischemic insult. Previously, we reported that the endogenous capacity of chicken hippocampal neurites to regenerate in response to HI injury was enhanced by GH treatment [15]. Also, in the adult rat, GH administration stimulated a significant increase in the dendritic length of cortical neurons [91]. However, the ability of GH to protect the length and number of neurites in pallial (cortical) neurons after HI damage had not been demonstrated until now. Our results clearly show that HI significantly reduced the number, length, and branching of neurites in the pallial cell cultures. In contrast, these morphometric parameters were clearly re-established by GH administration. Most remarkably, the co-incubation with specific neutralizing antibodies against cGH abolished the morphological effects induced by GH treatment, indicating that the modifications observed in the dendritic tree of pallial neurons correspond to a direct effect of GH. The mechanism by which GH stimulates neurite outgrowth could be associated with GAP-43 expression. GAP-43 plays an important role in dendritic growth and branching [92], which is essential for brain development. Previous studies showed that GAP-43 expression in rat hippocampus and cerebral cortex was increased after HI injury as a self-protection mechanism and was involved in the repair of damaged neurons [93]. Also, GH treatment was previously shown to increase GAP-43 in a hippocampal cell culture after HI injury [15]. Similarly, here we show that GH was able to increase GAP-43 mRNA expression in HI-damaged pallium. Moreover, GAP-43 overexpression has been associated with an increase in neurite outgrowth [94], which is consistent with GH effects increasing both GAP-43 and dendritic length, thus suggesting that GH facilitated neurite outgrowth in the cultured pallial injured neurons via this mechanism.

It has been reported that modification of neuron morphology is accompanied by synaptic modifications. So, we evaluated the effect of GH upon the expression of diverse synaptogenic markers. Neurexins 1 (NRXN1) and 3 (NRXN3) are synaptic cell-adhesion proteins that interact between pre- and postsynaptic neurons, and play essential roles in neurotransmission and the synapse [95]. It was found that NRXN1 and NRXN3 mRNA expression did not change under HI conditions alone, but was clearly increased by GH treatment in HI-damaged pallial cells. Furthermore, we found that the expression of neuroligin-1 mRNA (NLG1)—a transmembrane cell adhesion protein important for synapse development [96]—was significantly reduced by HI damage, but strongly upregulated after GH treatment in the injured cells. This is in accord with findings reported recently in hippocampal neurons [15]. Altogether, this evidence indicates that GH treatment exerts a stimulatory effect upon synaptic plasticity and thus protects neurons from HI damage.

Neurotrophins are growth factors that have an important role during brain development and are involved in processes that regulate plasticity, such as neural cell survival, renewal, differentiation, and axonal guidance, among others [97]. Local growth factors have been implicated in the endogenous mechanism of neuroprotection and neuroplasticity in response to brain injury [17,28,31]. Previous results in hippocampal cultures showed that after HI harm, the expression of local growth factors such as BDNF, NT-3, IGF-1, and BMP4 were increased [15], and this could explain the partial recovery observed in diverse parameters when the HI-injured hippocampal neurons were re-exposed to normal oxygenation conditions. Interestingly, GH addition to HI-injured cultures enhanced this endogenous recovery response, suggesting that GH acted together with local neurotrophins to repair the damage caused by HI. Here, we evaluated the local expression of BDNF, NT-3, IGF-1, BMP4, and GH mRNA in HI-harmed pallial cells. It was found that HI injury did not significantly affect local neurotrophin expression; however, the administration of GH treatment during HI injury importantly stimulated the expression of all the neurotrophins studied, including local GH mRNA. This result agrees with previous studies showing that endogenous expression of GH increased in cerebellum [53] and neural retina [73] after hypoxic and excitotoxic injuries, respectively, thus indicating that GH may play a critical role in the response mechanisms triggered by harmed neurons to preserve their survival and viability. It is known that BDNF and NT-3 are classical neurotrophins widely expressed in the cerebral cortex [98,99], and also that their administration induces brain neuroprotection and regeneration in several pathophysiological models [100]. In this study, we found that BDNF and NT-3 expression were significantly increased after GH treatment, which clearly correlated with the regenerative effect observed after HI injury. Furthermore, it has been described that circulating and locally expressed IGF-1—the classical mediator of GH actions in growth, homeostasis, and metabolism—is involved in neuroprotective actions in response to neural damage [101,102], and previous work from our group have indicated that some of the neurotrophic effects of GH could be mediated by IGF-1 [19,56]. Here, we demonstrated that GH induced a significant increase in IGF-1 mRNA expression after HI injury, which supports the notion that neuroprotective GH effects could be mediated, at least partially, by IGF-1. On the other hand, BMP4 is a neurotrophin that has been associated with the differentiation of neuronal stem cells (NSCs) [103,104]. Our results show that GH administration also induced a significant upregulation of BMP4 expression during HI injury; thus, it is possible that this neurotrophic factor acts together with GH to potentiate the proliferative effect observed in this study.

In conclusion, this work shows evidence that GH promoted protection and regeneration in the embryonic chicken brain pallium exposed to HI injury. GH promoted cell survival of HI-harmed pallial neurons, exerted a clear anti-apoptotic activity through the stimulation of ERK1/2 and Bcl-2 pathways, potentiated cell proliferation both in vitro and in vivo, promoted neurite outgrowth and branching, and induced synaptogenesis in cortical neurons after HI injury, probably through a joint action between its own effect and that of the locally expressed neurotrophic factors. This work provides additional evidence about the neuroprotective and regenerative effects of GH, and suggests its therapeutic potential as a neurotrophic factor that could be involved in combination treatments for HI encephalopathy and this possibility deserves further investigation.

## 4. Materials and Methods

### 4.1. Animals

The fertilized eggs used in this study (*Gallus gallus domesticus*, White Leghorn) were obtained from Pilgrim’s Pride (Querétaro, Mexico). The eggs were incubated at 39 °C in a humidified air chamber (IAMEX, Querétaro, Mexico), and rotated one-quarter of a revolution every 50 min during incubation. All experimental procedures were conducted according to national (NOM-062-ZOO-1999) and international (*The guide for care and use of laboratory animals*, U.S. National Research Council) guidelines, and were approved by the bioethical committee of the Instituto de Neurobiología, UNAM.

### 4.2. Primary Pallial Cell Cultures

Chick embryos at 9 days of embryogenesis (DE) were anesthetized in ovo by cooling them on ice for 5 min and then euthanized by decapitation. The brain was pulled out quickly from the developing calvarium and the meningeal tissue was peeled off. Later, the telencephalic pallia of both hemispheres were micro-dissected in a cold calcium/magnesium-free Hank’s balanced buffer using a microsurgical blade under a stereoscopic microscope. The pallial tissues from 12 embryos were digested in 0.002% trypsin–EDTA (Sigma-Aldrich, St. Louis, MO, USA) solution at 39 °C for 10 min, followed by mechanical dissociation with a glass pipette and passed through a 40 µm pore filter. Cells were counted in a Neubauer chamber using the trypan blue exclusion method, and then (2 × 10^6^) plated in 12-well plates (Costar Corning, NY, USA) or rounded 12 mm cover glasses coated with 50 µg/mL poly-L-lysine, and incubated in Neurobasal containing 2% B27, GlutaMAX, and 1% penicillin–streptomycin (Gibco, Grand Island, NY, USA) (Nb-B27). The cell cultures were stabilized in a humidified incubator at 39 °C, under an atmosphere of 95% air/5% CO_2_, for 3 days.

### 4.3. Treatments

#### 4.3.1. In Vitro Experiments

In vitro experimental hypoxic–ischemic (HI) conditions were induced by substituting normal Nb-B27 medium for DMEM—low glucose 1× (I g/L, Gibco, Grand Island, NY, USA) medium and pallial cell cultures were kept in a humidified hypoxic chamber (Napco E Series, Model 302 CO_2_ incubator) for 24 h at 37 °C and flushed with a 95% N_2_/5% CO_2_ gas mixture. The resulting oxygen concentration within the chamber (5% O_2_) was monitored with an ambient oxygen sensor (BW Technologies, Arlington, TX, USA) and maintained throughout the experiment. Immediately after this treatment, a group of cells were used as HI controls and analyzed. As the normoxic control group, cell cultures were maintained in normal Nb-B27 medium under normal aeration conditions (21% O_2_). In order to analyze whether GH was capable of protecting pallial cells from HI injury, the cultures were treated as follows: a) addition of 10 nM recombinant chicken GH (rcGH, Revholt, PRL, Israel) [15], or b) the combination of 10 nM rcGH with a 1:20 dilution of affinity-purified IgGs directed specifically against rcGH (α-GH) [105]. The treatments were administered to the cell cultures in DMEM—low glucose medium and remained during the 24 h of HI incubation conditions. At the end of this period, the cell cultures were submitted to several analyses.

In another set of experiments designed to evaluate whether GH was able to promote cell proliferation, a BrdU labeling assay was employed. For this, BrdU (10 µM) was diluted in either Nb-B27 medium or DMEM—low glucose medium and administered, together with the treatments, to the cell cultures of each experimental group (Normoxia, HI, and HI + GH).

#### 4.3.2. In Vivo Experiments

In vivo HI injury was induced by incubating fertilized chicken eggs from embryonic day 14 (ED14), in a humidified chamber at 37 °C (Napco E Series, Model 302 CO_2_ incubator) under a hypoxic atmosphere (12% O_2_) for 6 h. Immediately after this period, a group of eggs were used as HI controls and analyzed. As a normoxic control, a group of eggs were simultaneously incubated under normal conditions (21% O_2_). The experimental treatment groups were microinjected in ovo with either a single dose (2.25 µg) of rcGH (Revholt, PRL, Israel) diluted in 100 µL of injectable water or vehicle, just before the 6 h HI incubation. The corresponding analyses were performed immediately after this period. The effect of GH on cell proliferation was also evaluated in vivo. For this, GH (2.25 µg/dose) and BrdU (1.5 mg/dose) were diluted in 100 µL of injectable water. In ovo injections were administered using the following strategy: a first dose of GH + BrdU immediately before starting HI conditions and a second treatment with GH only at the end of the HI incubation. Then, the eggs were returned to the humidified air chamber and kept under normoxic conditions for another 24 h. After this additional period, the corresponding analyses were performed.

### 4.4. Determination of Cell Survival

Cell survival was assessed in the pallial cultures by the MTT (3-[4,5-dimethilthiazol-2-yl]-2,5-diphenyltetrazolium bromide) assay using a Vybrant kit (Molecular Probes, Eugene, OR, USA). After the treatments, culture media was substituted with DMEM medium without phenol red (Gibco, Grand Island, NY, USA), and the MTT reagent was added to each well (final concentration of 0.5 mg/mL) and then incubated for 4 h in a humidified atmosphere at 39 °C. The resulting formazan crystals were dissolved in 0.1 mL of solubilization solution (DMSO). Finally, the products were spectrophotometrically quantified at 570 nm in a microplate reader (Bio-Rad, Hercules, CA, USA).

### 4.5. Immunofluorescent Staining

The embryonic chicken brains were collected and post-fixed with 4% paraformaldehyde (PFA) for 48 h, and then immersed in 30% sucrose for an additional 24 h, both at 4 °C. Cryoprotected brains were cut into 40 µm thick coronal sections in a cryostat (Leica, CM3050S, Lake, IL, USA). On the other hand, cultured pallial cells (5 × 10^5^) were grown in round 12 mm cover glasses following the methodology previously described. Cell cultures were fixed with 4% paraformaldehyde for 20 min. Either brain sections or cell cultures were washed with TTBS (TBS + 0.05% Tween 20), 3 × 10 min. For BrdU staining, brain sections were treated with 2 M HCl at room temperature (RT) for 1 h, and cells for 20 min. To block non-specific signals, both fixed brain sections and cells were incubated with 5% non-fat milk (Bio-Rad, Hercules, CA, USA) in TBS for 45 min at room temperature (RT), and then washed in TTBS (3 × 10 min). Primary antibodies (Table 1) were diluted in 1% non-fat milk TTBS solution, added to the samples, and then incubated overnight at RT. For brain sections, BrdU primary antibody was incubated for 48 h at 4 °C. Then, brain sections and cell cultures were washed in TTBS (3 × 10 min), further incubated with the corresponding secondary antibodies (Table 1) for 2 h at RT, and then washed in TTBS (3 × 10 min). Later, the slides were incubated and counterstained with 300 nM DAPI (4′6-diamidino-2-phenylindole, Invitrogen) solution in TBS for 10 min and washed (3 × 10 min).

### 4.6. Cell Quantification

The hyperpallium (a chick brain region considered as homologous to the mammalian neocortex [106,107,108]), was analyzed using plates A 12.6–7.8 of the Atlas of Kuenzel and Masson [109]. Quantification of DCX+/DAPI cells was performed in six fields selected within the hyperpallium from six sections of each animal. For in vitro cell quantification, the total number of DAPI positive cells (DAPI+) and BrdU+/DAPI cells was estimated in at least 10 fields, in two different cover glasses per experiment, with a total of three independent experiments. Fluorescent immunolabeled sections and cover glasses were analyzed with an Olympus BX51 fluorescence microscope (Melville, NY, USA), employing a 20× objective. Images were acquired and reconstructed using FIJI-ImageJ software (https://fiji.sc, accessed on 30 November 2021).

### 4.7. SDS-PAGE and Western Blot Analysis

Cells or brain tissues were homogenized with an ultrasonicator (GE 130PB, Cole-Parmer, Vermon Hills, IL, USA) in 0.05 M HCl-Tris, pH 9.0 buffer, in the presence of a protease inhibitor cocktail (Mini-complete, Roche, Mannheim, Germany). The extracts were centrifuged (12,500 rpm, 15 min) and the supernatants collected. Total proteins were determined by the Bradford assay (Bio-Rad, Hercules, CA, USA), and 50 µg of protein/lane was separated in 12.5% SDS-PAGE slabs, and then transferred onto nitrocellulose membranes (Bio-Rad, Hercules, CA, USA) as previously described [53]. For immunoblotting, membranes were blocked with 5% non-fat milk (Bio-Rad) in TBS for 2 h. Then, membranes were incubated overnight at RT with the corresponding primary antibodies (Table 1) in TTBS. After washing the membranes with TTBS (3 × 10 min), they were incubated for 2 h with the corresponding HRP-conjugated secondary antibodies. Immunoreactive bands were visualized using ECL blotting detection reagent (Amersham-Pharmacia, Buckinghamshire, UK) on autoradiography film (Fujifilm, Tokyo, Japan). Kaleidoscope molecular weight markers (Bio-Rad) were used as reference for molecular mass determination.

### 4.8. Reverse Transcription Polymerase Chain Reaction (RT-PCR)

Total RNA was extracted from cells by adding 100 µL of TRIzol (Invitrogen, Waltham, MA, USA) according to the manufacturer’s indications. RNA was purified from cellular lysates using the Zymo Direct-zol purification kit (Zymo Research Corp., Irvine, CA, USA). Genomic DNA contamination was removed by DNAse I treatment (Invitrogen, Waltham, MA, USA) for 15 min at RT. The cDNA was synthetized from 2 µg of total RNA using 100 U of M-MLV reverse transcriptase (Promega, Madison, WI, USA), 1 mM dNTPs, 0.5 µg oligo d(T), and 0.5 µg random hexamers, for 60 min at 42 °C, in a final volume of 40 µL.

### 4.9. Quantitative PCR (qPCR)

The expression of target genes (Table 2) was quantified by qPCR using an ABI-PRISM 7900HT system (Applied Biosystems, Foster, CA, USA), using SYBR green (Maxima, Thermo Scientific, Waltham, MA, USA) in a 10 µL final volume containing 3 µL of diluted cDNA and 0.5 µL of each specific primer (0.5 µM). Primer sets used (Table 2) were designed to amplify avian mRNAs and to cross intron–exon boundaries to control for genomic DNA contamination. Reactions were performed under the following conditions: initial denaturation at 95 °C for 10 min, followed by 40 cycles of 95 °C for 15 s, 60 °C for 30 s, and 72 °C for 30 s. Dissociation curves were included after each qPCR experiment to ensure primer specificity. The relative abundance of the corresponding mRNAs was calculated using the comparative threshold cycle (Ct) method, employing the formula 2^−∆∆CT^ [110,111,112]. Gene expression determinations were performed in duplicate, from 4 independent experiments.

### 4.10. Determination of Apoptosis by Caspase-3 Activity

Apoptotic cell death in cell cultures was analyzed by using a caspase-3 colorimetric assay kit (Assay Designs Inc., Ann Arbor, MI, USA). The samples (8 µg protein) of cell lysates from each treatment, standards, p-nitroaniline (pNA) standard, and blank controls were placed in 96-well microplates in duplicate. After a 3 h incubation at 37 °C, the reaction was stopped with 1 N HCl, and absorbance at 405 nm was read immediately in a 3350-UV microplate reader (Bio-Rad, Hercules, CA, USA). The caspase-3 activity was calculated as units per microgram of protein [113], normalized and expressed as percent activity relative to the HI condition (which was considered as 100%).

### 4.11. Neurite Growth Analysis

After the treatments, cells were stained with the neuronal biomarker β-III-tubulin and visualized using an Olympus BX51 fluorescence microscope with a 40× objective. Images of individual neurons were captured with Lumenera software (Ottawa, ON, Canada). Neurites of each neuron (including all of its ramifications) were traced using Simple Neuro Tracer (Fiji freeware [114]), and total dendritic length and branch order analysis were automatically measured. For each group, 10 neurons from 8–10 randomly selected fields were analyzed, from three independent experiments.

### 4.12. Statistical Analysis

In all experiments, values were expressed as mean ± SEM. Significant differences between treatments were determined by one-way ANOVA with Tukey’s post hoc test. Unpaired Student’s *t* tests were used to compare between two groups where appropriate. Values of *p* < 0.05 were considered to be statistically significant.

## Figures and Tables

**Figure 1 ijms-23-09054-f001:**
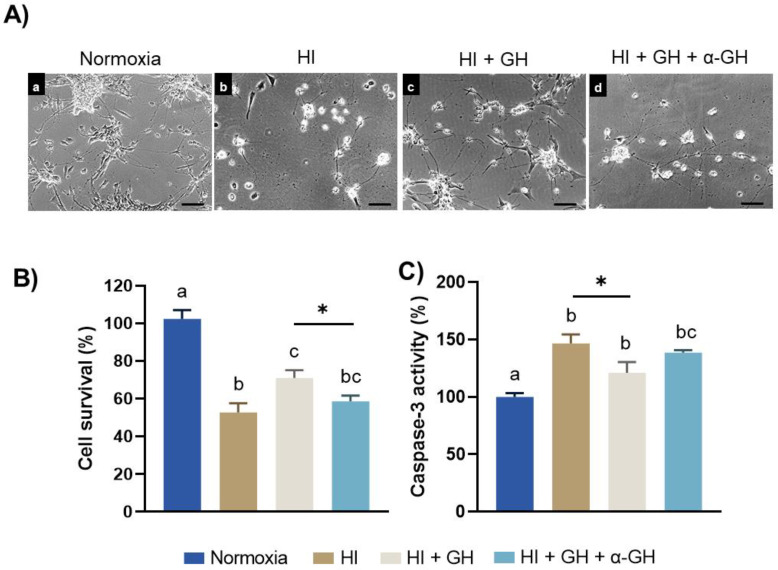
Effects of growth hormone (GH) upon cell survival and apoptosis in pallial primary cell cultures after a hypoxic–ischemic (HI) injury. (**A**) Representative phase–contrast images of primary cell cultures obtained from embryonic chicken pallium exposed to either: (a) normoxic, (b) HI, (c) HI + 10 nM GH, or (d) HI + 10 nM GH + 1:20 dilution of specific anti-cGH antibody (α-GH), incubation conditions. Scale bar: 50 µm. (**B**) Cell survival was evaluated by MTT assay, and (**C**) Caspase-3 activity was determined with a colorimetric assay in primary pallial cultures exposed to either normoxia, HI, HI + GH, and HI + GH + α-GH incubation conditions. Bars represent mean ± SEM (from 3 independent experiments in duplicate; *n* = 6 cultures per group). Groups with different letters are significantly different by one-way ANOVA with Tukey’s post hoc test (*p* < 0.05). An unpaired Student’s *t* test was used to compare HI versus HI + GH groups (panel **C**), and the asterisks (*) indicate a significant difference (* *p* < 0.05).

**Figure 2 ijms-23-09054-f002:**
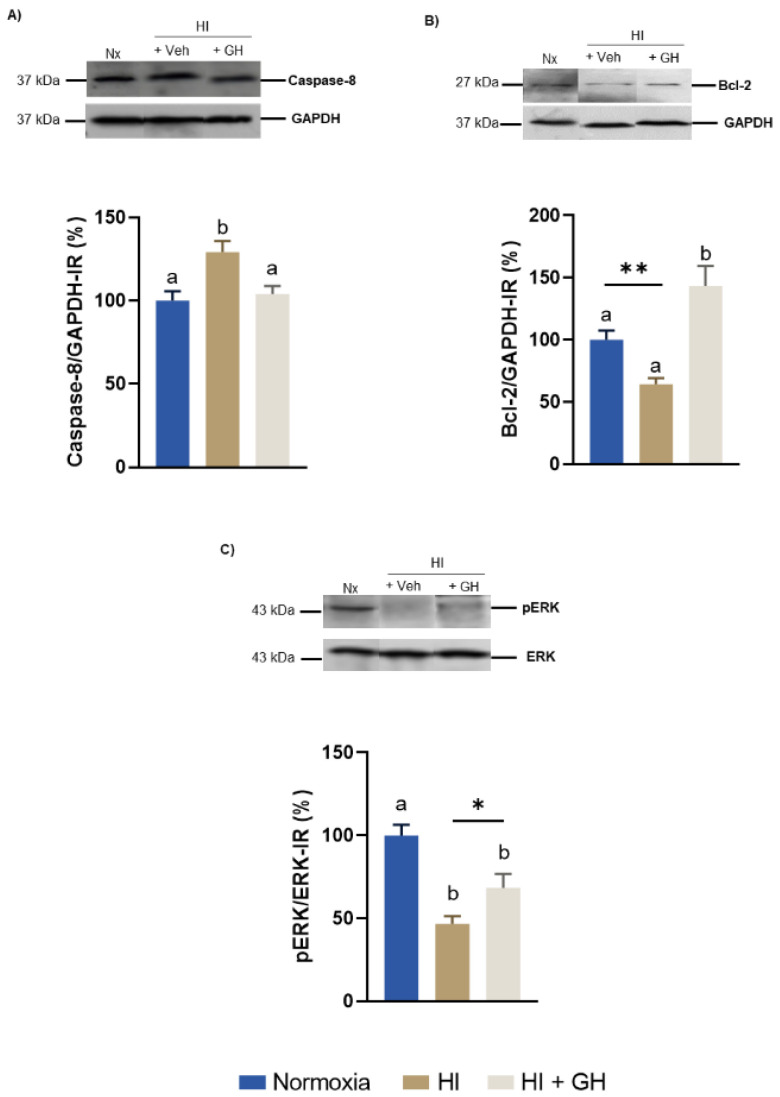
Effects of growth hormone (GH) upon modulation of apoptotic and pro-survival (caspase-8, Bcl-2, ERK1/2) pathways in the pallium of chick embryos exposed to a hypoxic–ischemic (HI) injury. (**A**) Densitometric analysis of immunoblots stained for Caspase-8 immunoreactivity-IR. (**B**) Densitometric analysis of immunoblots stained for Bcl-2 IR. In both cases, the values were corrected and normalized with GAPDH IR. (**C**) Densitometric analysis of immunoblots stained for pERK1/2 IR (corrected and normalized to total ERK1/2 IR). Bars represent mean ± SEM (*n* = 5 chick embryos per group). Groups with different letters are significantly different by one-way ANOVA with Tukey’s post hoc test (*p* < 0.01). An unpaired Student’s *t* test was used to compare normoxia versus HI + GH (panel **B**), and HI versus HI + GH groups (panel **C**). Asterisks (*) indicate significant differences (* *p* < 0.05, ** *p* < 0.001).

**Figure 3 ijms-23-09054-f003:**
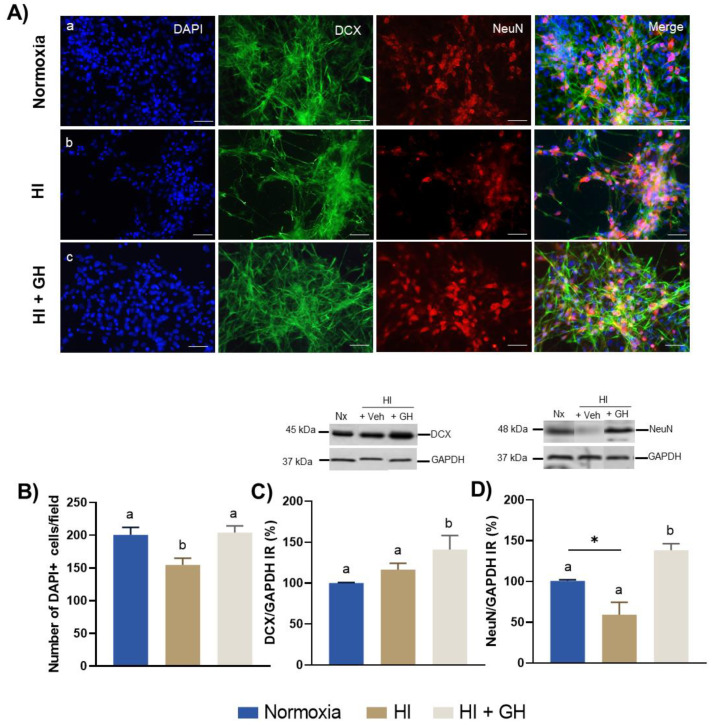
Growth hormone (GH) protects both immature and mature neuronal subpopulations after a hypoxic–ischemic (HI) injury in embryonic pallial cell cultures. (**A**) Representative immunofluorescence images showing effects upon cultures exposed to different incubation conditions: (**a**) normoxia, (**b**) HI, or (**c**) HI + 10 nM GH. Cell nuclei were stained with DAPI (blue), neuronal precursors with an antibody directed to doublecortin (anti-DCX; green), and mature neurons with an antibody directed to Neuronal Nuclear Antigen (anti-NeuN; red), and the images were merged. Scale bar: 50 µm. (**B**) Quantification of cell nuclei stained with DAPI in each experimental group. Bars represent mean ± SEM (from 3 independent experiments in duplicate; *n* = 6 cultures per group). (**C**) Representative Western blots and densitometric analysis of immunoblots stained for DCX IR. (**D**) Representative Western blots and densitometric analysis of immunoblots stained for NeuN IR. In both cases the values were corrected and normalized to GAPDH IR. Bars represent mean ± SEM (from 4 independent experiments; n = 4 cultures per group). Groups with different letters are significantly different by one-way ANOVA with Tukey’s post hoc test (*p* < 0.01). An unpaired Student’s *t* test was used to compare normoxia versus HI groups (panel **D**), and asterisks (*) indicate a significant difference (* *p* < 0.05).

**Figure 4 ijms-23-09054-f004:**
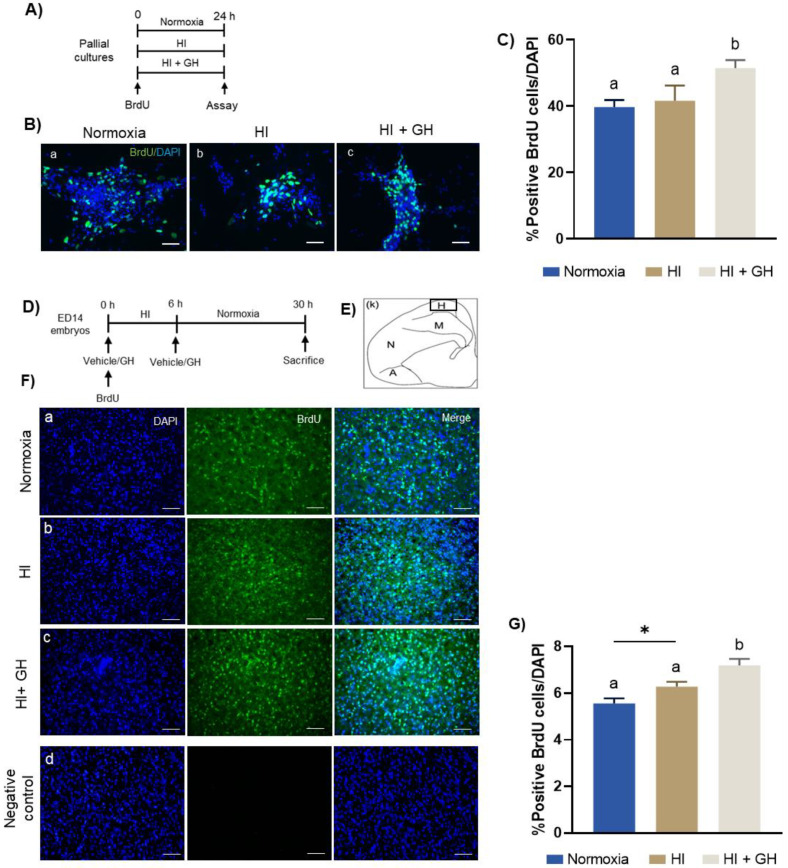
Growth hormone stimulates cell proliferation, both in vitro and in vivo, in embryonic chicken pallium exposed to a hypoxic–ischemic (HI) injury. (**A**) Schematic representation of time course and treatment protocol for the in vitro experiment. (**B**) Representative images of pallial neurons stained with BrdU (green) and DAPI (blue) in each experimental condition: (**a**) normoxia, (**b**) HI, (**c**) HI + 10 nM GH. Scale bar: 50 µm. (**C**) Relative proportion (%) of positive BrdU cells, expressed in relation to total DAPI labeled cells under each experimental condition. Bars represent mean ± SEM (3 independent experiments in duplicate; *n* = 6 cultures per group). Groups with different letters are significantly different by one-way ANOVA with Tukey’s post hoc test (*p* < 0.05). (**D**) Schematic representation of time course and treatment protocol for the in vivo experiment. (**E**) Representative drawing of a coronal section of the embryonic chick brain showing the location of the hyperpallium (**H**) region. (**F**) Representative immunofluorescence images showing effects of the treatments upon BrdU+ cells in the hyperpallium of the chick embryos exposed to: (**a**) normoxia, (**b**) HI, or (**c**) HI + 10 nM GH. Cell nuclei were stained with DAPI (blue) and BrdU with a specific antibody (green), and the images were merged. (**d**) Negative controls without anti-BrdU primary antibody. (**G**) Positive BrdU cells, quantified as relative percentage in relation to DAPI labeled cells under each experimental condition. Bars represent mean ± SEM (*n* = 4 chick embryos per group, 6 fields were quantified per hyperpallium/embryo). Groups with different letters are significantly different by one-way ANOVA with Tukey’s post hoc test (*p* < 0.0001). An unpaired Student’s *t* test was used to compare normoxia versus HI groups (panel **G**), and asterisks (*) indicate a significant difference (* *p* < 0.05).

**Figure 5 ijms-23-09054-f005:**
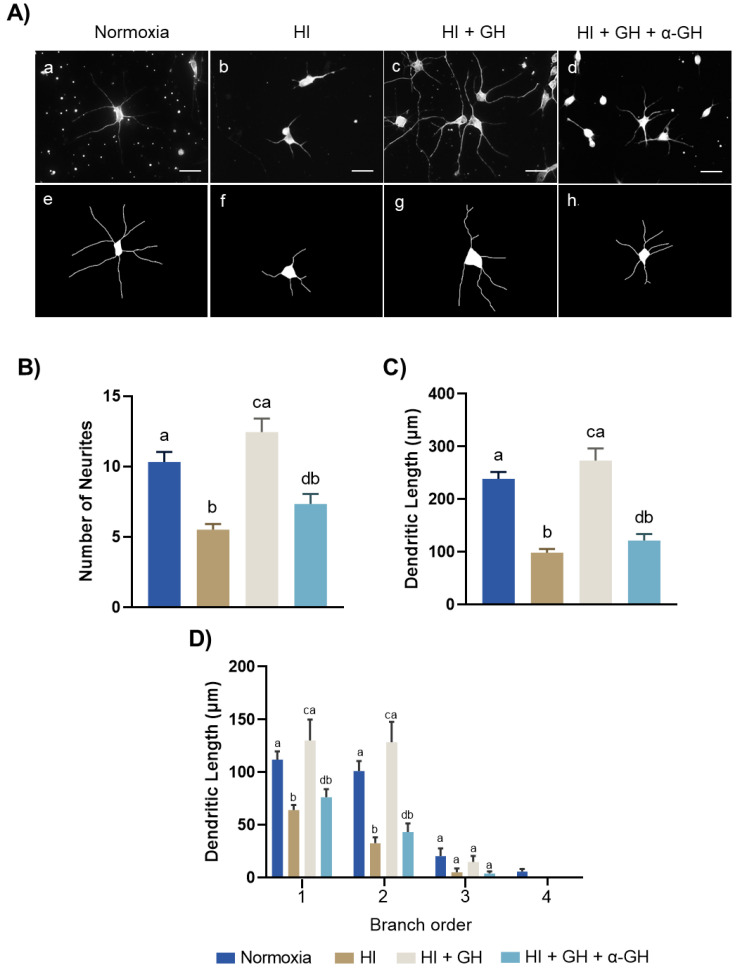
Effect of growth hormone (GH) upon neurite outgrowth after a hypoxic–ischemic (HI) injury. (**A**) Representative images of pallial neurons stained with β-III-tubulin under each experimental condition: (a) normoxia, (b) HI, (c) HI + 10 nM GH, (d) HI + 10 nM GH + 1:20 dilution of specific anti-cGH antibody (α-GH). Scale bar: 50 µm. Schematic drawings of neurons stained with β-III-tubulin used for morphometric analysis, in: (e) normoxia, (f) HI, (g) HI + GH, (h) HI + GH + α-GH. (**B**) The number of neurites was evaluated in 10 individual neurons for each group. (**C**) For dendritic length analysis, individual neurons were selected and drawn using the Simple Neuro Tracer plug-in of FIJI-ImageJ software. (**D**) Branching dendritic analysis realized on pallial neurons under each experimental condition. Bars represent mean ± SEM (from 3 independent experiments in duplicate; *n* = 6 cultures per group). Groups with different letters are significantly different by one-way ANOVA with Tukey’s post hoc test (*p* < 0.05).

**Figure 6 ijms-23-09054-f006:**
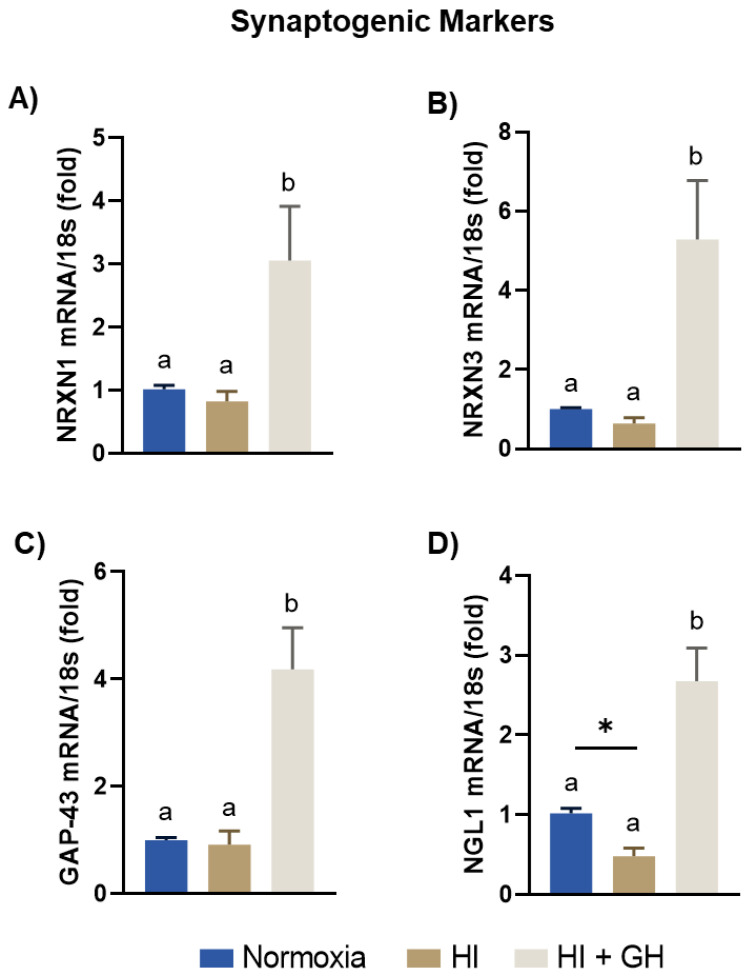
Effects of growth hormone (GH) upon the expression of synaptogenic markers after exposing pallial cultures to HI injury. Panels show the expression of: (**A**) NRXN1, (**B**) NRXN3, (**C**) GAP-43, and (**D**) NGL1 mRNAs, determined by RT-qPCR, under each incubation condition (normoxia, HI, or HI + GH). Relative mRNA expression values were corrected by the comparative threshold cycle (Ct) method and employing the formula 2^−ΔΔCT^. Ribosomal 18s RNA was used as the housekeeping gene. Bars represent mean ± SEM (from 4 independent experiments in duplicate; *n* = 8 cultures per group). Groups with different letters are significantly different by one-way ANOVA with Tukey’s post hoc test (*p* < 0.05). An unpaired Student’s *t* test was used to compare normoxia versus HI (panel **D**), and asterisks (*) indicate a significant difference (* *p* < 0.05).

**Figure 7 ijms-23-09054-f007:**
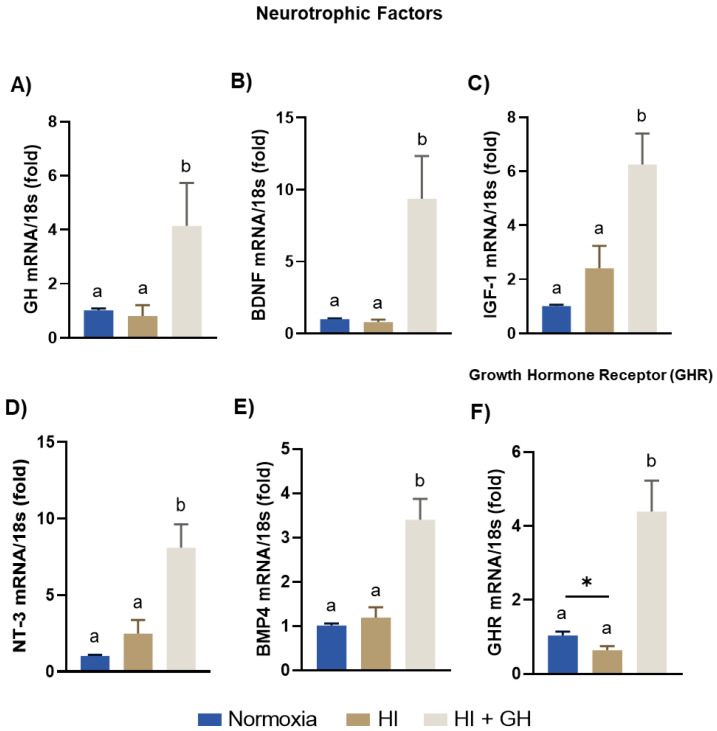
Effects of growth hormone (GH) upon endogenous neurotrophins and GHR expression after exposing pallial cultures to HI injury. Panels show the expression of: (**A**) GH, (**B**) BDNF, (**C**) IGF-1, (**D**) NT-3, (**E**) BMP4, and (**F**) GHR mRNA, determined by RT-qPCR, under each incubation condition (normoxia, HI, HI + GH). Relative mRNA expression values were corrected by the comparative threshold cycle (Ct) method and employing the formula 2^−ΔΔCT^. Ribosomal 18s RNA was used as the housekeeping gene. Bars represent mean ± SEM (from 4 independent experiments in duplicate; *n* = 8 cultures per group). Groups with different letters are significantly different by one-way ANOVA with Tukey’s post hoc test (*p* < 0.05). An unpaired Student’s *t* test was used to compare normoxia vs. HI (panel **F**), and asterisks (*) indicate a significant difference (* *p* < 0.05).

**Table 1 ijms-23-09054-t001:** Antibodies.

Target	Host/Type	Dilution	Source	Cat. No.
DCX	Guinea pig/polyclonal	1:250 (IF) 1:1000 (WB)	Millipore	AB2253
NeuN	Mouse/monoclonal	1:250 (IF) 1:1000 (WB)	Millipore	MAB377
BrdU	Rat/monoclonal	1:50	Abcam	AB-6326
β-III-tubulin	Mouse/monoclonal	1:250	Abcam	Ab78078
HIF-1 α	Rabbit monoclonal	1:1000	Abcam	Ab2185
Caspase-8	Rabbit/polyclonal	1:500	Santa Cruz	SC-6134
Bcl-2	Rabbit/polyclonal	1:500	Santa Cruz	SC-492
p44/42 MAPK (Erk1/2)	Mouse/monoclonal	1:1000	Cell signaling	91065
P-p44/42 MAPK (Erk1/2)	Mouse/monoclonal	1:1000	Cell signaling	4370S
GAPDH	Rabbit/monoclonal	1:3000	Cell signaling	14C10
Guinea pig IgG	Goat/Alexa fluor 488	1:1000	Invitrogen	A-11073
Mouse IgG	Goat/Alexa fluor 595	1:1000	Invitrogen	A-11032
Rat IgG	Goat/Alexa fluor 488	1:1000	Invitrogen	A-11029
Guinea pig IgG	Goat/HRPconjugated	1:3000	Millipore	AP108P
Mouse IgG	Goat/HRPconjugated	1:3000	Abcam	AB-20043
Rabbit IgG	Goat/HRPconjugated	1:3000	Invitrogen	A-65611

**Table 2 ijms-23-09054-t002:** Oligonucleotides.

Target	Primer	Sequence (5′-3′)	Size	Accesion #
cGH	Fwd Rev	CGCACCTATATTCCGGAGGAC GGCAGCTCCATGTCTGACT	128 bp	NM_204359
cBDNF	Fwd Rev	AGCAGTCAAGTGCCTTTGGA TCCGCTGCTGTTACCCACTCG	167 bp	NM_001031616
cIGF1	Fwd Rev	TACCTTGGCCTGTGTTTGCT CCCTTGTGGTGTAAGCGTCT	170 bp	NM_001004384
cNT3	Fwd Rev	AGGCAGCAGAGACGCTACAAC AGCACAGTTACCTGGTGTCCT	248 bp	NM_001109762
cBMP4	Fwd Rev	CGCTGGGAGACCTTTGATGT CCCCTGAGGTAAAGATCGGC	153 bp	NM_205237.3
cGHR	Fwd Rev	ACTTCACCATGGACAATGCCTA GGGGTTTCTGCCATTGAAGCTC	180 bp	NM_001001293.1
cNRXN1	Fwd Rev	CCACTCTGATCATTGACCGGG CGCCAGACCTTCCACATAGT	392 bp	NM_001198975.1
cNRXN3	Fwd Rev	GCTGGGTCTCTCTTTGGGTC CACCCACAA AAAGGTCGCTG	394 bp	NM_001271923.1
cNLG1	Fwd Rev	CTCCAGTGTGTCCCCAGA AC CATCACAGGCTTAGGTCCCC	170 bp	NM_001081502.1
cGAP43	Fwd Rev	AGGAGCCTAAACAAGCCGAC TGCTGGGCACTTTCAGTAGG	178 bp	NM_001305054.1
C18s	Fwd Rev	CTCTTTCTCGATTCCGTGGGT TTAGCATGCCAGAGTCTCGT	100 bp	M59389

## Data Availability

The data that support the findings of this study are available from the corresponding author upon reasonable request.

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
