# Peer review of "Neuroprotective and Regenerative Effects of Growth Hormone (GH) in the Embryonic Chicken Cerebral Pallium Exposed to Hypoxic–Ischemic (HI) Injury"

_ijms, 2022, doi:10.3390/ijms23169054_

Round 1

Reviewer 1 Report

The authors undertook interesting studies to check whether growth hormone has a neuroprotective and regenerative effect in the brain cortex.

The authors made an in-depth review of the literature on the basis of which they wrote the introduction to the article. The research was properly planned. The research methods used were well described. The assumed research objectives have been achieved. The obtained research results have been correctly interpreted. The statistical analysis of the results is correct. Based on the obtained results, the authors formulated correct conclusions. For this reason, the work may be recommended for publication in its current form.

Author Response

Reviewer 1.

We express our gratitude to Reviewer 1 for the positive and supportive opinion about our study and the results we report in this manuscript, regarding the neuroprotective and regenerative effects of growth hormone in response to hypoxia in the chicken brain pallium. We also very much appreciate his/her recommendation to publish it in its current form.

Reviewer 2 Report

The paper describes the protective effects of GH on HI injury 

The paper describes the effects of GH on HI of Embryonic Chicken Cerebral Pallium. I have the following comments:

1. In Fig. 1B , Fig. 2, Fig. 3, Fig. 4, Fig. 6 the effects of the GH antibody is not shown. These data should be provided.

2. It would be much more informative to the reader to show directly the degree of hypoxia-induced metabolic rewiring by HIF-1alpha related effects (either directly in WB or indirectly by up-regulation of target genes like SLC2A1 etc.).

3. The potential interaction of GH-signaling and HIF1alpha signaling should be discussed.

Author Response

Reviewer 2.

Thanks to Reviewer 2 for his/her comments about our manuscript and suggestions to improve it.

Following are specific responses to each query raised by the referee:

  1. In Fig. 1B, Fig. 2, Fig. 3, Fig. 4, Fig. 6 the effects of the GH antibody is not shown. These data should be provided.

In the revised version of the manuscript, we have now modified Figure 1B to include the effect of adding the anti-GH antibody (1:20 dilution) upon cell survival in primary pallial cultures exposed to hypoxic-ischemic (HI) injury while also incubating in the presence of 10 nM GH. As can be observed, the antibody significantly decreased the stimulatory effect provoked by GH in response to HI. (lines 104-106). The Figure legend was modified accordingly (lines 114-115).

This result is consistent with others previously shown in the manuscript (Fig. 1Ad, Fig. 5Ad, Fig. 5Ah, Fig. 5B, Fig. 5C, Fig. 5D), regarding the specific capacity of the anti-GH antibody to decrease or neutralize the stimulatory effect of GH upon cell number and survival, as well as the number, length and branching of neurites, in pallial cultures exposed to HI. Overall, these results indicate that the neuroprotective effects observed in response to neural injury involve directly the action of GH.

Also, the blocking effect of the anti-GH antibody upon cell survival under HI in our study is consistent with the report that GH knockdown (using a specific siRNA) significantly reduced cell viability and increased necrosis in chicken cerebellar cultures exposed to an acute hypoxic-ischemic injury (Baltazar-Lara et al., Int. J. Mol. Sci. 2021, 22, 256).

The Referee asks to include the effects of adding the anti-GH antibody in Figs. 2, 3, 4, and 6. However, this would need to repeat again all those experiments with the appropriate number of replicates, and that is clearly beyond the time frame given to submit the revised version of the manuscript to the Journal. We think there is sufficient evidence of the specific blocking effect of the GH antibody upon the neuroprotective action of GH in the several results shown here.

Since we are continuing our studies to further understand the mechanisms involved in GH neuroprotective effects during HI injury in the brain, we will include the addition of anti-GH antibody or the transfection of specific GH siRNA to analyze the effects of blocking either externally administered or locally expressed GH, respectively; and the results will be reported in future papers.

  1. It would be much more informative to the reader to show directly the degree of hypoxia-induced metabolic rewiring by HIF-1alpha related effects (either directly in WB or indirectly by up-regulation of target genes like SLC2A 1 etc.).

We thank the Reviewer for this suggestion to analyze what happens with the HIF-1a pathway in our model of HI injury in the embryonic chicken pallium and determine if there are any interactions (and if so, which) with the GH effects described in this study.

As it has been described, hypoxia triggers the stabilization of hypoxia-inducible factor 1 (HIF-1a) subunit in the cytoplasm, which then translocate to the nucleus and dimerize with HIF-B and other coactivators to form a functional HIF transcription factor. This, in turn, binds to hypoxia-responsive elements (HRE) in the promoter regions of target genes involved in cellular adaptation events to hypoxia, to improve cell survival (v.gr. protective, trophic or angiogenic, such as VEGF, EPO, iNOS); and regulate energetic metabolism (e.g. glucose transport, glycolytic enzymes, among others).

We have now included a supplementary figure (S1 and legend) showing that effectively HIF-1a significantly increased in the pallial neural tissue (130.4 ± 15.2%, p <0.05) when the chicken embryos were exposed to hypoxia, as compared to normoxic conditions (100 ± 3.88%), as determined by Western blot (mentioned in lines 123-128). This result is consistent with a previous report where we showed that HIF-1a was augmented in embryonic chicken cerebellar cultures (Baltazar-Lara et al., Int. J. Mol. Sci. 2021, 22, 256), by immunocytochemistry. These data indicate that, in our hypoxic-ischemic models, both in vivo and in vitro, the hypoxic conditions employed trigger the activation of HIF-1a.

Although we agree with the Reviewer that it would be very interesting to show which intracellular mechanisms related to HIF-1a are turned on in our HI paradigm, and how they interact with the GH effects described in this manuscript, we think that this would be the matter of a separate study.

Such a new investigation could include the analysis of GH effector pathways (v.gr. JAK/STAT, PI3K/Akt, PKA, mTOR) upon HIF-1a transcription and/or post-translational stabilization (hydroxylation, ubiquitination), under both normoxic and hypoxic conditions. Also, to determine and compare their effects (HIF-1a and GH) on the expression of several markers associated with the early and late responses to hypoxia (e.g. VEGF, EPO, iNOS, GLUT1, glycolytic enzymes, etc.). But this clearly corresponds to another study.

  1. The potential interaction of GH-signaling and HIF1alpha signaling should be discussed.

As mentioned above, during a hypoxic and ischemic event, the stability and half-life of the transcription factor HIF-1a increases, it binds to the beta subunit and translocate to the nucleus to regulate the expression of genes whose function is to increase oxygen delivery (angiogenesis and regulation of vascular tone), reduce ROS and modulate oxygen consumption.

On the other hand, it is known that GHR activation involves the participation of several signal pathways, such as JAK/STAT, PI3K/AKT, and MAPK. The multiactivity of these pathways increases the complexity in the communication among neural cells.

It has been reported that the transcription factors of STAT family promote the expression of the Hif1a gene. Therefore, it is possible that during a hypoxic event, GH (in addition to stimulate neuroprotection) can promote and potentiate the expression and stabilization of HIF-1a, as well as the effects of this transcription factor on energetic metabolism and cell survival.

We have now added a paragraph in the Discussion section regarding these issues (lines 378-399), with its corresponding references [61-70]. Because of this, the number of references grew to 116.

Round 2

Reviewer 2 Report

The authors have addressed my concerns accordingly.